# A Systematic Review of Machine-Translation-Assisted Language Learning for Sustainable Education

**Xinjie Deng** and **Zhonggen Yu** *

Faculty of Foreign Studies, Beijing Language and Culture University, Beijing 100083, China; 202121198711@stu.blcu.edu.cn
* Correspondence: yuzhonggen@blcu.edu.cn

**Abstract:** With the rapid development of artificial intelligence, machine translation (MT) has gained popularity in recent years. This study aims to present a systematic review of literature on MT-assisted language learning in terms of main users, theoretical frameworks, users' attitudes, and the ways in which MT tools are integrated with language teaching and learning. To this end, relevant peer-reviewed articles ($n$ = 26) were selected through the Preferred Reporting Items for Systematic Review and Meta-Analysis Protocol (PRISMA-P) for further analysis. The findings revealed that the main MT users were undergraduate and graduate students. Both teachers and students held mixed attitudes for different reasons. It was also found that MT integration followed four steps, i.e., introduction, demonstration, task assignment, and reflection. The procedures of MT integration could be updated and perfected by introducing other features in the future.

**Keywords:** machine translation (MT); MT tools; language learning; users' attitudes; systematic literature review

## 1. Introduction

With the rapid development of artificial intelligence, machine translation (MT) has gained popularity in recent years. MT tools are widely used among individuals, from pupils [1] to senior citizens [2]. In everyday life, this technology can be used for household tasks, news, personal communication, internet shopping, entertainment, and legal services [2]. Most importantly, the 21st century has witnessed much integration of MT into education, especially in language learning. MT has been implemented in classroom practice at different levels, ranging from primary education [3] to tertiary education [4]. Naturally, there is growing interest in the ways that MT tools can be implemented to facilitate students' learning.

The impact of MT implementation varies with language-learning skills. MT has been found to be effective in promoting students' writing skills. Lee [5] compared two versions of students' writing, i.e., the initial English-as-a-second-language writing and the revised English writing with the assistance of machine translation. Through revisions, students significantly increased their writing scores and reduced lexical and grammatical errors. MT could also improve writing quality regarding syntactic complexity and accuracy [6]. However, MT tools may not satisfactorily improve translation skills if students overly rely on them. The penalty points of students' translations with the help of machine translation were significantly higher than those of translations without MT involvement [7].

The mixed results of MT integration have necessitated further exploration of machine-translation-assisted language learning (MTALL) research. Most students want to receive training and strategies for effective MT use from their teachers [8]. With the goal of sustainable foreign-language education, this review study aims to examine the main users of MT tools, theoretical frameworks, MT users' attitudes, and the existing methods of MT integration. It is expected to provide references for researchers and practitioners to explore effective ways of integrating MT in language teaching and learning in the future.

Some review studies on MTALL have been conducted. For instance, these include a systematic review focused on the number of publications, MT quality, users' perceptions, and the effectiveness of MT [9]. Current development and practices of MT in translation teaching have also been explored [10]. However, Kanglang and Afzaal [10] did not specify databases and search strategies. Another review study analyzed the publication trends, methodologies, research themes, and the types of MT [11]. The above studies were different from the present study, which concentrated on some other critical aspects of sustaining language education, as shown in Table 1.

**Table 1.** The comparison between previous studies and this study.

| N. | Study | Databases | Quality Assessment | Main Users | Theoretical Frameworks | Users' Perceptions | The Ways of MT Integration |
|---|---|---|---|---|---|---|---|
| 1 | [9] | Cambridge Core, Science Direct, JSTOR, ProQuest, EBSCO, Google Scholar, and six journals | × | × | × | √ | × |
| 2 | [10] | Not available | × | × | × | × | × |
| 3 | [11] | Eight journals indexed by SSCI and CSSCI | × | × | × | × | × |
| 4 | This study | WoS Core Collection, Sage, Wiley, ERIC, and EBSCO | √ | √ | √ | √ | √ |

## 2. Literature Review

The 21st century has witnessed digital transformation of language education. The transformation has helped students to reach recent digital tools and improve efficiency [12]. Artificial intelligence (AI), the leading technology in digital transformation [13], has been employed in language teaching and learning. For example, virtual reality and augmented reality tools can improve language learning by immersing students into virtual worlds [14]. AI chatbots served as conversation partners in language-speaking classes [15]. Among the emerging technologies, machine translation (MT) also transformed existing teaching methods.

Machine translation (MT) boasts a long history, dating back to the 1940s [16]. It refers to the automatic process by which written or spoken texts are translated from one natural language to another using computer software and applications [8,17]. MT has three types, i.e., rule-based machine translation (RBMT), statistical machine translation (SMT), and neural machine translation (NMT). The advent of NMT and the release of the NMT system by Google in 2016 dramatically improved the translation quality [4]. Since then, the number of translation tools integrated with NMT has surged.

MT has contributed to various educational fields, including nursing, science, and language. Nursing users have benefited from MT systems since it has enabled them to easily understand professional articles across the world [18]. In university science courses, it is not uncommon for biology and microbiology students to study bilingually using MT tools [19]. Moreover, MT as a pedagogical tool is instrumental in second-language writing [5], and the MT evaluation metrics are considered effective to assess language learners' translations and interpretations [20]. In digital collaborative writing, students can encounter problems when interacting using MT tools [21]. Thus, students should develop their MT literacy when they use MT as an aid for English academic writing [22]. The MT literacy that Bowker [22] emphasized comprises six elements, i.e., privacy, academic integrity, potential for algorithmic bias, awareness of different tools, awareness of different translation tasks, and improving the output by changing the input. In foreign language reading, creativity in translation may be an important factor to improve students' reading engagement [23].

Compared to an increasing number of studies on MTALL, attempts to synthesize the related studies are limited. Among the few reviews, Zhen et al. [11] examined 40 articles from SSCI- and CSSCI-indexed journals. However, the limitation was the scope that the included studies only came from high-impact journals. Moreover, they did not explore the

main users of MT tools. Another recent study conducted by Lee [9] systematically reviewed MTALL studies from 2000 to 2019 in terms of publication trends, users' perceptions, and learning outcomes. Nevertheless, neither of them focused on the theoretical frameworks used in MTALL research. The authors thus proposed the following research questions:

RQ1. Who are the main users of MT tools?
RQ2. What are the frequently used theoretical frameworks adopted in MTALL research?

Users' attitudes towards educational technologies play a significant role in technology implementation. Teachers' unwillingness to explore digital technologies could influence their mindsets and teaching practices, impeding technology integration in language teaching [24]. However, positive attitudes towards technology use contribute to the strong intention to adopt technology, even for preservice teachers [25]. The same goes for language learners. Students recognizing the benefits of greater technology use possessed a higher level of learning motivation, thus facilitating students' technology-based self-directed learning [26]. Although students have been found to hold mixed attitudes towards MT [9], the review article only included studies published before 2020. Kanglang and Afzaal [10] also called for more exploration of the challenges of MT in translation teaching, which deserves further investigation in terms of users' attitudes. The authors presented the following research question:

RQ3. What are the users' attitudes towards machine translation tools?

MT integration in language teaching and learning has received academic attention. The statistical machine translation syllabus was an effective curriculum design for translation students [27]. However, the teaching design in Tian's [28] study failed. It was designed to use MT as a self-editing tool for intermediate Chinese language learners. Thus, it is necessary to review the recent literature and explore the ways in which previous studies integrated MT tools. Furthermore, the perceived benefits and drawbacks of MT applications highly depended on the quality of MT integration [29]. MT has brought about considerable changes, and the number of MTAL publications has increased in recent years [9]. The authors thus proposed the following research question:

RQ4. How are MT tools integrated with language teaching and learning?

## 3. Research Methods

This review study implemented rapid evidence assessment of the literature method [30] based on the Preferred Reporting Items for Systematic Review and Meta-Analysis Protocol (PRISMA-P). However, the review has not been registered yet. Firstly, the researchers conducted a literature search on several online databases based on the proposed research questions. Secondly, they screened the retrieved studies based on inclusion and exclusion criteria. Thirdly, the researchers identified the included studies by using a quality assessment questionnaire. Lastly, the researchers examined and synthesized the literature. Figure 1 shows the selection process of the retrieved studies. The PRISMA checklist could be found in Supplementary Materials.

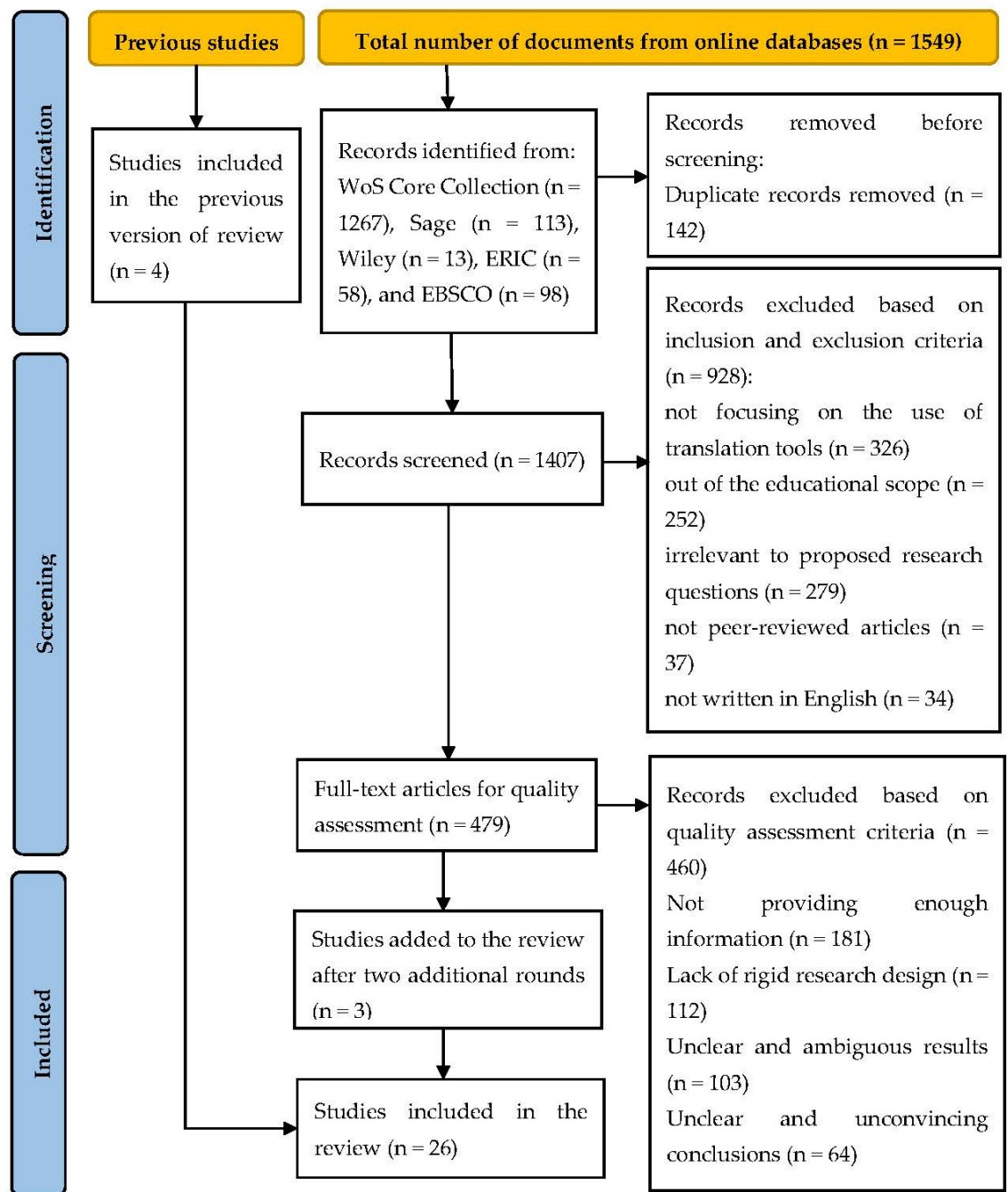

**Figure 1.** A flow diagram of literature selection.

### 3.1. Literature Search

The researchers developed the search strategy and obtained literature by searching digital databases. On 7 May 2022, they employed Boolean operators and retrieved 1267 results from the Web of Science (WoS) Core Collection by keying in "translation tool*" OR "translation engine*" OR "machine translation" OR "Google Translate" OR "Bing Translate" (topic) and learn* OR teach* OR student* OR educat* (topic) and language OR "foreign language" OR "second language" (topic). The researchers also searched literature from Sage, Wiley, the Education Resources Information Center (ERIC), and EBSCO (Academica Search Premier and Business Source Premier). The time ranged from January 2012 to 7 May 2022.

The researchers obtained 113 results from Sage and 13 results from Wiley by keying in "translation tool OR translation engine OR machine translation OR Google Translate OR Bing Translate" in the title and "learn OR teach OR education OR student OR lan-

guage OR foreign language OR second language" in the abstract. They also retrieved 58 results from ERIC and 98 results from EBSCO by keying in (title: "translation tool*" OR "translation engine*" OR "machine translation" OR "Google Translate" OR "Bing Translate") AND (abstract: learn* OR teach* OR educat* OR student) AND (abstract: language OR foreign language OR second language). Therefore, the literature search generated 1549 results in total, which were imported to EndNote in the BibTex format to remove the duplicated studies.

### 3.2. Inclusion and Exclusion Criteria

The researchers then selected the studies based on inclusion and exclusion criteria. The studies were included if they (1) focused on the educational use of translation tools; (2) targeted at least one research question in this review study; (3) were peer-reviewed journal articles; and (4) were written in English. The studies were excluded if they (1) did not focus on the use of translation tools; (2) were out of the educational scope; (3) were irrelevant to the research questions in this review study; (4) were not peer-reviewed articles; and (5) were not written in English.

### 3.3. Quality Assessment

To ensure the quality of the finalized studies, the researchers conducted the quality assessment based on the University of West England Framework [31] to refine the results. They evaluated each study using the following questionnaire:

(a) The study provided enough information for this review study. The options were "Yes (2)", "Limited (1)", and "No (0)".
(b) The study was rigidly designed, and the research design was clearly described. The possible answers were "Yes (2)", "Limited (1)", and "No (0)".
(c) The presentation of the results was clear and unambiguous. The possible answers were "Yes (2)", "Limited (1)", and "No (0)".
(d) The study arrived at clear and convincing conclusions. The options were "Yes (2)", "Limited (1)", and "No (0)".

Therefore, the score range of this assessment questionnaire was between 0 and 8. Two researchers scored the literature with high inter-rater reliability ($k = 0.83$). The researchers included 26 studies with ratings of seven and more in the finalized studies. Appendix A presents the basic information and quality assessment scores of each included study.

## 4. Results

### 4.1. RQ1. Who Are the Main Users of MT Tools?

Table 2 provides information on the main users of MT tools. The major group was undergraduate and graduate students, since a total of 17 out of 26 studies were conducted among them. Elementary school students, secondary school students, and university educators received less attention, with two studies each. Most researchers showed the least interest in other types of participants, including pre-university students, elementary school teachers, and preservice teachers. Moreover, there were 22 studies targeting students, whereas only four studies focused on teachers, indicating that students attracted much more attention than teachers and educators.

Some other detailed information is also worth noticing. Two studies did not clearly explain the types of users. The study conducted by Nino [32] only specified students' languages and levels, while the other study [33] reported the participants' language background, age, gender, and learning experience. In addition, it was found that the total number of the studies classified by user types exceeded the number of included studies, as two studies involved mixed users. Both graduate students and university teachers participated in Hellmich and Vinall's study [34], while Kelly and Hou [1] interviewed elementary school students and teachers.

**Table 2.** Main users of machine-translation (MT) tools.

| N. | Main Users | Included Studies | Total Number |
|---|---|---|---|
| 1 | Elementary school students | [1,35] | 2 |
| 2 | Secondary school students | [6,36] | 2 |
| 3 | Preuniversity students | [37] | 1 |
| 4 | Undergraduate and graduate students | [4,5,7,27,34,38–49] | 17 |
| 5 | Elementary school teachers | [1] | 1 |
| 6 | University educators | [34,50] | 2 |
| 7 | Preservice teachers | [29] | 1 |
| 8 | Not available | [32,33] | 2 |

*4.2. RQ2. What Are the Frequently Used Theoretical Frameworks Adopted in MTALL Research?*

Less than half of the included studies (N = 9) adopted theoretical frameworks, including seven theories, models, and frameworks. These were the taxonomy of error types, CALF (syntactic complexity, accuracy, lexical complexity, and fluency) measures, translanguaging, the ecological theoretical framework, the technology acceptance model, the TPACK (technological, pedagogical, and content knowledge) framework, and the ADAPT (amending, discussing, assessing, practicing, and training) approach.

The taxonomy of error types used in Groves and Mundt's study [37] was proposed by Ferris et al. [51]. This taxonomy focused on detailed errors and was designed for assessing the linguistic accuracy of the translation [37]. There were twenty prominent error types in this taxonomy, ranging from punctuation errors to sentence-structure errors. Groves and Mundt [37] adopted this taxonomy to analyze the grammatical errors in the English text produced by Google Translate. They found that word choice, sentence structure, and missing words were the three most common types of error.

CALF measures are the most widely used methodological framework to measure linguistic performance. They included measures in syntactic complexity, accuracy, lexical complexity, and fluency [48]. CALF measures could provide information about language improvement and writing quality, thus contributing to many researchers choosing these measures in their studies (e.g., [52]). Chung and Ahn [48] employed a CALF framework to examine the effect of machine-translation tools on multidimensional aspects of Korean students' English writing. They also investigated CALF measures across different text genres and proficiency levels.

Translanguaging is a dominant theory in multilingual education and also a pedagogical approach [53]. It highlights that linguistic practices are fluid and that languages are integrated linguistic systems used for communicative purposes [54]. Thus, it is also considered to be the pedagogy that encourages students to motivate all linguistic resources for learning and understanding. Kelly and Hou [1] utilized a translanguaging framework to explore how pupils and teachers perceive machine translation in multilingual learning. This theory can be appropriately used to appraise machine translation and understand bilingual or multilingual students' uses of it.

The ecological approaches to translation education take teaching and learning as ecosystems. Multiple ecological components may interact across multiple levels within a larger ecology [55]. Thus, from an ecological perspective, researchers are likely to consider essential components in an educational setting, as they may influence pedagogical practices. Hellmich and Vinall [34] analyzed foreign language instructors' beliefs about machine translation through an ecological lens. Following the ecological theoretical frame, they extended not only data-collection measures to closed-ended and open-ended surveys, but also the areas of inquiry into students' uses and motivations, and the teaching profession.

Since machine-translation tools are closely associated with technology, the technology acceptance model (TAM) is also widely used. This model stemmed from the theory of reasoned action [56] and was developed by Davis [57]. It is usually used to examine individuals' acceptance of a new technology. The TAM involves three key components, i.e., perceived usefulness, perceived ease of use, and behavioral intention. Yang and

Mustafa [41] focused on perceived usefulness and perceived ease of use to investigate Chinese undergraduate students' responses to postediting of machine translation. Furthermore, some studies adapted or extended the model by introducing other factors. For example, Tsai and Liao [44] developed an adapted TAM to understand the relationships among perceived usefulness, perceived ease of use, students' English learning motivation, and reading anxiety. Integrating students' experience in using machine translation and learning motivation, Yang and Wang [45] also developed an extended model to explore students' intention to use machine translation.

The last two frameworks primarily target teachers and instructors. The technological, pedagogical, and content knowledge (TPACK) framework originated from Shulman [58], who proposed utilizing teachers' pedagogical and content knowledge. With the rapid development of technology, Mishra and Koehler [59] emphasized the importance of technological knowledge in teaching. Ross et al. [29] utilized the TPACK framework to discuss how preservice teachers integrate translation tools into their lessons. Likewise, the ADAPT approach adopted by Knowles [33] was also designed for the integration of machine translation. The five elements of this approach are amending assignments, discussing translation tools, assessing with translation tools in mind, practicing integrity, and training learners to use translation tools.

### 4.3. RQ3. What Are the Users' Attitudes towards Machine Translation Tools?

In the finalized studies, more than half (N = 16) explored students' attitudes, while only a minority of studies (N = 4) focused on teachers' perspectives. Most students and teachers held positive attitudes towards machine translation, while there remained some challenges relating to this technology.

### 4.3.1. Students' Attitudes

Students reported several benefits of machine translation, which played different roles in actual pedagogical use. It served as an essential survival tool for pupils to communicate with their teachers and peers [1]. Many studies also indicated that machine translation acted as a language learning resource [35,38]. It was instrumental for students in gaining knowledge, including vocabulary, grammar, pronunciation, and spelling [4,27,33,38]. Considered as a writing assistant, the translation tool improved students' translation quality and writing quality, thus boosting their confidence [4,39,47].

Machine translation, per se, boasts many characteristics. It has been deemed easy to use [41], efficient [43,48], and reliable [7,39]. Machine translation has proven to be more effective and accurate for word selection than students' lexical choices from the dictionary [5]. Furthermore, in other studies, students favored five writing modes [38] and the speech function possessed by most translation tools [36]. Students could choose any mode, e.g., photo taking, handwriting, or letter typing, to write on and carry out the application [38].

However, some students were skeptical about machine translation and raised their concerns. Students felt machine outputs to be unreliable and inaccurate in terms of idioms and phrases [39], sentence structures [41,48], grammar [5,47–49], certain writing styles [5,43], and contexts [48]. These resulted in little trust in machine translation. On the other hand, students may become overdependent on the machine output and accept it without postediting [33,35], suggesting a lack of critical analytic skills [40]. In this case, they would develop bad habits and become less motivated and enthusiastic to learn [39]. Occasionally, translation tools failed to identify languages because they do not take a standardized form, e.g., in the case of Tagalog [35]. Technical issues [41,42] and cheating [33,36] were also responsible for students' negative attitudes.

### 4.3.2. Teachers' Attitudes

Teachers take positive attitudes towards machine translation as they have experienced the benefits of introducing this technology in the classroom. Through a translation applica-

tion, teachers could communicate with dual language learners in students' home languages, thus achieving positive interactions and providing students with a sense of security [1,29]. Teachers preferred to regard machine translation as a multimodal tool to help beginners to understand instructions and participate in the class [1]. A majority of language instructors did not believe that machine translation threatened their profession. Instead, they were expected to integrate this technology into classroom practices [34].

Despite general support for machine translation, a few teachers did not favor it. They treated this technology as a hindrance to the development of translation skills, since students might get accustomed to it [50]. Teachers were also worried about the translation of complete sentences and whole paragraphs [34]. Other reasons included lack of confidence and organizational support [29]. Thus, teachers held that they should receive formal training in machine translation [50]. Moreover, it was time-consuming to translate instructions into different languages [1].

### 4.4. RQ4. How Are MT Tools Integrated with Language Teaching and Learning?

Some integration procedures can be identified when teachers attempt to incorporate MT into their curriculum, as presented in Figure 2. Language teachers firstly presented and introduced MT tools to students by identifying the role of these tools in class and constructing affordances of them [35], or by discussing with students the associated ethical issues, advantages, and pitfalls [4]. After the introduction, teachers did demonstrations and instructed students in the use of MT tools [7,29,35]. Ensuring that students were capable of utilizing translation tools, teachers could instruct students to complete tasks and assignments individually [46] or collaboratively [42]. MT tools could also be integrated with interesting classroom activities to help students interact with peers [35]. The last procedure involved reflection on MT integration [29].

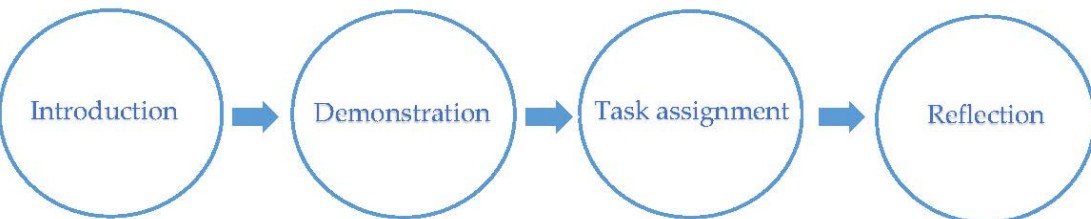

**Figure 2.** Four steps of MT integration.

The tasks assigned by teachers were primarily in the form of writing (e.g., [35,39,48]). In most cases, students were asked to write an essay with a given title in their first language (L1) (e.g., [5,46,47,49]). After that, students compared the self-translated L2 text and the machine-translated L2 text. However, there were some exceptions. After finishing L2 writing tasks, students translated their drafts into L1 with the help of MT tools. The L1 version was then translated into L2 by MT tools. Thus, students made comparisons between self-written L2 versions and machine-translated L2 versions [4]. In Cancino and Panes' [6] study, each of three groups wrote a story in their second language (L2) based on given images. Two experimental groups could turn to MT for help. Groves and Mundt [37] used Google Translate to directly translate students' essays from their L1 to L2, analyzing errors made by the MT tool.

Role play and translation tasks also appear in language teaching and learning. Through role play where one preservice teacher acted as a student, and the other as a teacher [29], the preservice teachers could become familiar with MT applications and troubleshoot the potential problems they may encounter in actual teaching. Students in a group played different roles, e.g., leader, terminologist, documentation specialist, translator, and editor, to translate passages together using MT tools, e.g., Wordfast Anywhere [42]. Moreover, the experimental group in Olkhovska and Frolova's study [7] also individually translated the text from English into Ukrainian with the assistance of an MT engine. The translation

materials were presented in different forms, including reading texts, audio recording extracts, and video recording extracts [32].

Reflection from both teachers and students can benefit the improvement of MT-assisted language learning. Students reflected critically on technical and evaluation issues [27], confidentiality, ethical issues, and pricing policies [43]. Reflection papers helped students to realize their own perceptions and attitudes [5]. Teachers were encouraged to reflect on the uses, successes, and difficulties of MT integration in class [29]. Specifically, Ross et al. [29] posed a list of reflective questions for teachers in terms of MT incorporation, successful parts, teaching behaviors, teaching impact, students' responses, and modifications.

## 5. Discussion

This review systematically analyzed previous studies on machine translation from the perspective of main users, theoretical frameworks, learning performance, and users' attitudes. RQ1 aimed to explore the main users of machine translation in teaching and learning. The findings showed that previous studies focused more on students than teachers and educators, especially tertiary education students. This was probably because students could easily get access to machine-translation systems [40]. Another possible reason was that most researchers, working at universities and colleges, adopted the convenience-sampling technique for their studies (e.g., [45,49]).

RQ2 attempted to identify theoretical frameworks used for MTAL research. However, many studies did not use any theory or model as the basis for the research. Researchers used the taxonomy of error types and CALF measures since both provided various aspects to analyze written texts [48]. Translanguaging and the ecological theoretical framework were two new perspectives used to understand machine translation in education, while the TPACK framework and the ADAPT approach were two practical frameworks helping teachers to integrate machine translation into educational settings [29,33]. Most importantly, the TAM or the extended TAM were the most widely used frameworks mainly because they can consider both intrinsic features of machine-translation use and external factors associated with learning.

RQ3 focused on students' and teachers' attitudes towards MT tools. Both students and teachers held mixed attitudes with different reasons. Students treated MT tools as learning resources but disfavored them because of technical defects and learners' overdependence. When students were required to finish assignments, MT tools seemed to reduce their academic pressure but simultaneously impede the cultivation of their creative and critical thinking. However, it may be demanding for students alone to achieve the balance. From the perspective of teachers, MT tools served as teaching assistants, though some teachers felt it challenging to manipulate them and introduce them into classroom teaching. The mixed feelings of users were likely to stem from the advantages and drawbacks of MT per se and the ways in which users apply the technology.

RQ4 tried to understand how MT tools are incorporated in language teaching and learning. Teachers and researchers usually followed four steps, i.e., introduction, demonstration, task assignment, and reflection. The main tasks were writing tasks, translation tasks, and role play. The designs of writing tasks varied from study to study. Furthermore, reflections on MT integration covered many aspects, which were closely related to students' and teachers' attitudes.

The greatest difference among task designs was which method language students should use to write essays. The rationale behind writing in L1 first might be reducing cognitive load and maximizing efficiency [60]. It is believed that L1 and L2 are naturally linked to each other in the writers' minds [61]. Nevertheless, writing in L1 first was questioned due to the inhibition of writing in the target language. The other choice, writing in L2 first, may stimulate learners' motivation to write in a target language [3]. However, this task design required students to either write in L2 with the aid of MT or translate their L2 writings into L1 and use MT to compare the self-written L2 version and the MT L2

version. The former design could not be used make comparisons, while the latter design involved translation twice.

## 6. Conclusions

This conclusion section consists of the major findings, limitations of this study, and future research directions.

### 6.1. Major Findings

This review study aimed to provide an analysis of previous studies on MT-assisted language learning in terms of main users, theoretical frameworks, users' attitudes, and the ways that MT tools are integrated with language teaching and learning. The findings showed that the main MT users were undergraduate and graduate students. Teachers and students held mixed attitudes for different reasons. It was also found that MT integration followed four steps, i.e., introduction, demonstration, task assignment, and reflection.

### 6.2. Limitations

There are several limitations in this study. In the first place, the study may not include all related publications due to the limitation of library resources. In the second place, the procedures of MT integration may vary with advances in educational technologies. In the third place, there may be other MT integration patterns used for language teaching and learning, which deserve further exploration.

### 6.3. Implications for Future Research

One of the major findings of this study was that both teachers and students held mixed attitudes towards using MT tools. Researchers could further investigate the factors contributing to users' mixed attitudes by extending the existing models. Future research could also adopt diverse theoretical frameworks and consider interdisciplinary research methods. Teachers should be aware that their beliefs and perceptions of technology use may influence their teaching practice [24]. They could also pay attention to their students' attitudes towards the design of MT integration and make adjustments. In addition, future research could focus on primary and secondary school students and teachers.

With the development of information technology, the procedures of MT integration could be updated and perfected by introducing other features. For example, mobile learning technologies and social media tools may play critical roles in foreign language learning [62]. The adoption of gamification could also improve and sustain online learning [63]. Future research could also compare the effectiveness of various task designs on students' learning achievements. Teachers and course designers are encouraged to explore appropriate ways of introducing MT tools that are pertinent to teaching content. Educational institutes are expected to provide training for teachers and students in order to improve their digital literacy skills [64] and effectively integrate MT into foreign language education.

In the future, researchers could examine how to systematically reflect on the design of MT integration. They could consider learners' strategies and teachers' guidance [46] since both may influence students' learning outcomes. They may also investigate gender differences [65] in MT-assisted language learning. Course developers should establish objectives of MT-integrated curriculum and the corresponding evaluation systems. Teachers could utilize learning tools, e.g., assessment cards and questionnaires, to guide students to truthfully give their feedback on the courses.

**Supplementary Materials:** The following supporting information can be downloaded at: https://www.mdpi.com/article/10.3390/su14137598/s1, File S1: PRISMA_2020_checklist.

**Author Contributions:** Conceptualization, X.D. and Z.Y.; methodology, X.D.; software, X.D.; validation, X.D. and Z.Y.; formal analysis, X.D. and Z.Y.; investigation, X.D. and Z.Y.; resources, X.D.; data curation, X.D.; writing—original draft preparation, X.D.; writing—review and editing, Z.Y.; visualiza-

tion, X.D.; supervision, Z.Y.; project administration, Z.Y.; funding acquisition, Z.Y. All authors have read and agreed to the published version of the manuscript.

**Funding:** This research was funded by the 2019 MOOC of Beijing Language and Culture University (MOOC201902) (Important) "Introduction to Linguistics"; the "Introduction to Linguistics" of online and offline mixed courses in Beijing Language and Culture University in 2020; the special fund of the Beijing co-construction project—research and reform of the "Undergraduate Teaching Reform and Innovation Project" of Beijing higher education in 2020—an innovative "multilingual+" excellent talent training system (202010032003); and the research project of the graduate students of Beijing Language and Culture University, "Xi Jinping: The Governance of China" (SJTS202108).

**Institutional Review Board Statement:** Not applicable.

**Informed Consent Statement:** Not applicable.

**Data Availability Statement:** Not applicable.

**Acknowledgments:** The authors would like to extend their gratitude to the people who helped in this study and the projects which financially supported this study.

**Conflicts of Interest:** The authors declare no conflict of interest.

## Appendix A

**Table A1.** The quality assessment of included studies.

| N. | Study | Theoretical Framework | Research Instruments | Applications or Platforms | Research Foci | Quality Assessment | | | | |
|---|---|---|---|---|---|---|---|---|---|---|
| | | | | | | (a) | (b) | (c) | (d) | Total |
| 1 | [37] | The taxonomy of error types | Students' essays | Google Translate | The linguistic accuracy of English translation product | 1 | 2 | 2 | 2 | 7 |
| 2 | [32] | Not available | Tasks and students' oral reflections | Not available | Students' attitudes towards MT | 1 | 2 | 2 | 2 | 7 |
| 3 | [4] | Not available | Tasks and questionnaires | Google Translate, Baidu Translate, and Sogou Translate | Students' attitudes towards MT | 1 | 2 | 2 | 2 | 7 |
| 4 | [50] | Not available | Questionnaires | Not available | Translation educators' attitudes towards MT | 1 | 2 | 2 | 2 | 7 |
| 5 | [27] | Not available | Questionnaires and students' reflections | SmartMATE | Students' perceptions of MT syllabus and self-evaluation of learning outcomes | 1 | 2 | 2 | 2 | 7 |
| 6 | [29] | The TPACK framework | Teachers' reflections | Speak & Translate | Teachers' attitudes towards MT | 2 | 2 | 2 | 2 | 8 |
| 7 | [38] | Not available | Interviews | Google Translate | Learners' perceived affordances of the application and experiences with it | 1 | 2 | 2 | 2 | 7 |
| 8 | [39] | Not available | Interviews | Google Translate | Learners' behaviors and attitudes towards MT use | 1 | 2 | 2 | 2 | 7 |
| 9 | [40] | Not available | Questionnaires | Google Translate and Naver Translate | Students' use of MT and attitudes towards it | 1 | 2 | 2 | 2 | 7 |
| 10 | [41] | Technology acceptance model | Questionnaires | Google Translate, Bing Translate, and Baidu Translate | Students' responses to postediting of MT tools | 2 | 2 | 2 | 2 | 8 |

**Table A1.** *Cont.*

| N. | Study | Theoretical Framework | Research Instruments | Applications or Platforms | Research Foci | Quality Assessment | | | | |
|---|---|---|---|---|---|---|---|---|---|---|
| | | | | | | (a) | (b) | (c) | (d) | Total |
| 11 | [42] | Not available | Tests | Wordfast Anywhere | The effect of the translation tool on students' translation skills | 1 | 2 | 2 | 2 | 7 |
| 12 | [5] | Not available | Writing tasks, interviews, and reflection papers | Google Translate and Papago | The effect of translation tools on students' English writing and students' attitudes towards translation tools | 2 | 2 | 2 | 2 | 8 |
| 13 | [35] | Not available | Observations | Google Translate | The ways in which students use the translation tool | 1 | 2 | 2 | 2 | 7 |
| 14 | [43] | Not available | Questionnaires | Apertium, Systran, DeepL, Google, Translate 2018, MemSource, and MateCat | Students' attitudes towards the use of translation tools | 1 | 2 | 2 | 2 | 7 |
| 15 | [1] | Translanguaging | Pupil focus groups and teachers' interviews | Not available | Students' and teachers' attitudes towards MT | 2 | 2 | 2 | 2 | 8 |
| 16 | [7] | Not available | Tests | Microsoft Translator | The effect of machine translation on students' translation quality | 1 | 2 | 2 | 2 | 7 |
| 17 | [44] | Technology acceptance model | Questionnaires and structural equation analysis | Not available | Students' behavioural learning patterns in MT use | 2 | 2 | 2 | 2 | 8 |
| 18 | [45] | Technology acceptance model | Questionnaires | Not available | Students' intention to use MT by considering experience and motivation | 2 | 2 | 2 | 2 | 8 |
| 19 | [6] | Not available | Writing tasks | Google Translate | The effects of the translation tool on writing quality | 1 | 2 | 2 | 2 | 7 |
| 20 | [36] | Not available | Posts and responses from Student Room forum | Google Translate | Students' attitudes towards the translation tool use | 1 | 2 | 2 | 2 | 7 |
| 21 | [46] | Not available | Evaluators' scores and reflections | Google Translate | The comparison between students' translation and machine translation and factors influencing machine translation quality | 1 | 2 | 2 | 2 | 7 |
| 22 | [47] | Not available | Writing tasks and questionnaires | Google Translate | The effect of the translation tool on English writing and students' attitudes towards Google Translate | 1 | 2 | 2 | 2 | 7 |
| 23 | [48] | CALF measures | Writing tasks and questionnaires | Google Translate | The effect of the translation tool on linguistic features and students' attitudes toward Google Translate | 2 | 2 | 2 | 2 | 8 |
| 24 | [49] | Not available | Writing tasks and questionnaires | Google Translate | The comparison between students' translation and machine translation and students' attitudes towards Google Translate | 2 | 2 | 2 | 2 | 8 |

**Table A1.** *Cont.*

| N. | Study | Theoretical Framework | Research Instruments | Applications or Platforms | Research Foci | Quality Assessment | | | | |
|---|---|---|---|---|---|---|---|---|---|---|
| | | | | | | (a) | (b) | (c) | (d) | Total |
| 25 | [34] | Ecological theoretical framework | Questionnaires | Not available | Foreign language instructors' attitudes towards MT | 2 | 2 | 2 | 2 | 8 |
| 26 | [33] | The ADAPT approach | Questionnaires | Google Translate | Students' attitudes towards the translation tool | 2 | 2 | 2 | 2 | 8 |

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
