# Peer review of "A Systematic Review of Machine-Translation-Assisted Language Learning for Sustainable Education"

_sustainability, doi:10.3390/su14137598_

Round 1

Reviewer 1 Report

A Systematic Review of Machine Translation-Assisted Language Learning for Sustainable Education

  This paper presents a systematic review of literature on machine translation (MT) assisted language learning in terms of main users, theoretical frameworks, users’ attitudes, and the way the relevant tools are integrated with the educational process. 

The review used 26 relevant peer-reviewed articles selected via the Preferred Reporting Items for Systematic Review and Meta-analysis Protocol (PRISMA-P). The findings revealed that main MT the identity of the users (teacher, undergraduate and graduate students) which possess mixed attitudes for various reasons. In addition it was found that the integration process of MT follows four steps, i.e., introduction, demonstration, task assignment, and reflection. The procedures of MT integration could be updated 16 and perfected by introducing other features in the future

Comments;

 This kind of paper, that is, a systematic review of literature, is evaluated with different criteria than a paper presenting empirical research where one could demand theoretical underpinnings and methodological robustness. I will use just three criteria, here, which are: an elucidated methodological approach, the comprehensible presentation and the usefulness to the field.

  Reading this paper, I found that the paper, given the above criteria, meets the underling standards. The paper is well written and following the known PRISMA-P methodology for systematic review of literature. The steps followed are adequately described and the finding are clearly presented.

Of course a meta-analysis could include and additional quantitative elaboration of the data, however, I do not want to raise other issue concerning data analysis and I will stay with what the authors intended to present in this paper.

 Given that the field of research is novel, a systematic review is undoubtedly very useful to the researchers and the benefits of such contribution is obvious.

Reviewer 2 Report

Dear author

I had a good time reading your research paper. The idea is interesting and would encourage this to be bought to light. However, this paper would need more editing and reviewing. Do not lose hope. You are almost there!

The literature review as well as the structure of the paper and flow of ideas were all well done! However, there was a lack of literature on the topic of what machine translation contributed to the education field, and since this is a part of content in Digital Transformation of Education section, it would be relevant to include.

The results could have been more accurate and sharp, but a weak introduction, methodology and literature review have minimized the importance of the results found.

The idea of this research paper is interesting and one to be thoroughly read/analyzed. However, there need to be a lot more work done in bridging the gap between theory and practice and what is the unique of current study compared to the previous work. This is mainly because the existing research on the variable of digital transformation process in education sector, language learning, was not thoroughly mentioned in the literature review.

Nonetheless, this idea is interesting and unique. I would encourage the author to do more research and include data, tests, graphs, regression models and so on.

The idea of this research paper is interesting, however, there needs to be a lot more information, literature review and data involved in relations to the independent variable. I would suggest using other publications from this journal, in order to get an idea of the structure and type of research information to include.

Have a good job effort for paper improvement.
